# wav2vec 2.0: A Framework for Self-Supervised Learning of Speech Representations

**Alexei Baevski**  **Henry Zhou**  **Abdelrahman Mohamed**  **Michael Auli**

{abaevski,henryzhou7,abdo,michaelauli}@fb.com

**Facebook AI**

## Abstract

We show for the first time that learning powerful representations from speech audio alone followed by fine-tuning on transcribed speech can outperform the best semi-supervised methods while being conceptually simpler. wav2vec 2.0 masks the speech input in the latent space and solves a contrastive task defined over a quantization of the latent representations which are jointly learned. Experiments using all labeled data of Librispeech achieve 1.8/3.3 WER on the clean/other test sets. When lowering the amount of labeled data to one hour, wav2vec 2.0 outperforms the previous state of the art on the 100 hour subset while using 100 times less labeled data. Using just ten minutes of labeled data and pre-training on 53k hours of unlabeled data still achieves 4.8/8.2 WER. This demonstrates the feasibility of speech recognition with limited amounts of labeled data.[1]

## 1 Introduction

Neural networks benefit from large quantities of labeled training data. However, in many settings labeled data is much harder to come by than unlabeled data: current speech recognition systems require thousands of hours of transcribed speech to reach acceptable performance which is not available for the vast majority of the nearly 7,000 languages spoken worldwide [31]. Learning purely from labeled examples does not resemble language acquisition in humans: infants learn language by listening to adults around them - a process that requires learning good representations of speech.

In machine learning, self-supervised learning has emerged as a paradigm to learn general data representations from unlabeled examples and to fine-tune the model on labeled data. This has been particularly successful for natural language processing [43, 45, 9] and is an active research area for computer vision [20, 2, 36, 19, 6].

In this paper, we present a framework for self-supervised learning of representations from raw audio data. Our approach encodes speech audio via a multi-layer convolutional neural network and then masks spans of the resulting latent speech representations [26, 56], similar to masked language modeling [9]. The latent representations are fed to a Transformer network to build contextualized representations and the model is trained via a contrastive task where the true latent is to be distinguished from distractors [54, 49, 48, 28] (§ 2).

As part of training, we learn discrete speech units [53, 32, 7, 18] via a gumbel softmax [24, 5] to represent the latent representations in the contrastive task (Figure 1) which we find to be more effective than non-quantized targets. After pre-training on unlabeled speech, the model is fine-tuned

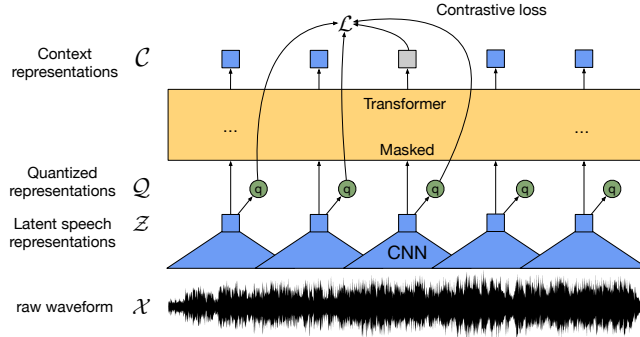

Figure 1: Illustration of our framework which jointly learns contextualized speech representations and an inventory of discretized speech units.

on labeled data with a Connectionist Temporal Classification (CTC) loss [14, 4] to be used for downstream speech recognition tasks (§ 3)

Previous work learned a quantization of the data followed by a contextualized representations with a self-attention model [5, 4], whereas our approach solves both problems end-to-end. Masking parts of the input with Transformer networks for speech has been explored [4, 26], but prior work relies either on a two-step pipeline or their model is trained by reconstructing the filter bank input features. Other related work includes learning representations from auto-encoding the input data [52, 11] or directly predicting future timesteps [8].

Our results show that jointly learning discrete speech units with contextualized representations achieves substantially better results than fixed units learned in a prior step [4]. We also demonstrate the feasibility of ultra-low resource speech recognition: when using only 10 minutes of labeled data, our approach achieves word error rate (WER) 4.8/8.2 on the clean/other test sets of Librispeech. We set a new state of the art on TIMIT phoneme recognition as well as the 100 hour clean subset of Librispeech. Moreover, when we lower the amount of labeled data to just one hour, we still outperform the previous state of the art self-training method of [42] while using 100 times less labeled data and the same amount of unlabeled data. When we use all 960 hours of labeled data from Librispeech, then our model achieves 1.8/3.3 WER (§ 4, § 5).

## 2 Model

Our model is composed of a multi-layer convolutional feature encoder $f : \mathcal{X} \mapsto \mathcal{Z}$ which takes as input raw audio $\mathcal{X}$ and outputs latent speech representations $\mathbf{z}_1, \ldots, \mathbf{z}_T$ for $T$ time-steps. They are then fed to a Transformer $g : \mathcal{Z} \mapsto \mathcal{C}$ to build representations $\mathbf{c}_1, \ldots, \mathbf{c}_T$ capturing information from the entire sequence [9, 5, 4]. The output of the feature encoder is discretized to $\mathbf{q}_t$ with a quantization module $\mathcal{Z} \mapsto \mathcal{Q}$ to represent the targets (Figure 1) in the self-supervised objective (§ 3.2). Compared to vq-wav2vec [5], our model builds context representations over continuous speech representations and self-attention captures dependencies over the entire sequence of latent representations end-to-end.

**Feature encoder.** The encoder consists of several blocks containing a temporal convolution followed by layer normalization [1] and a GELU activation function [21]. The raw waveform input to the encoder is normalized to zero mean and unit variance. The total stride of the encoder determines the number of time-steps $T$ which are input to the Transformer (§ 4.2).

**Contextualized representations with Transformers.** The output of the feature encoder is fed to a context network which follows the Transformer architecture [55, 9, 33]. Instead of fixed positional embeddings which encode absolute positional information, we use a convolutional layer similar to [37, 4, 57] which acts as relative positional embedding. We add the output of the convolution followed by a GELU to the inputs and then apply layer normalization.

**Quantization module.** For self-supervised training we discretize the output of the feature encoder $\mathbf{z}$ to a finite set of speech representations via product quantization [25]. This choice led to good

results in prior work which learned discrete units in a first step followed by learning contextualized representations [5]. Product quantization amounts to choosing quantized representations from multiple codebooks and concatenating them. Given $G$ codebooks, or groups, with $V$ entries $e \in \mathbb{R}^{V \times d/G}$, we choose one entry from each codebook and concatenate the resulting vectors $e_1, \ldots, e_G$ and apply a linear transformation $\mathbb{R}^d \mapsto \mathbb{R}^f$ to obtain $\mathbf{q} \in \mathbb{R}^f$.

The Gumbel softmax enables choosing discrete codebook entries in a fully differentiable way [16, 24, 35]. We use the straight-through estimator [26] and setup $G$ hard Gumbel softmax operations [24]. The feature encoder output $\mathbf{z}$ is mapped to $\mathbf{l} \in \mathbb{R}^{G \times V}$ logits and the probabilities for choosing the $v$-th codebook entry for group $g$ are

$$p_{g,v} = \frac{\exp(l_{g,v} + n_v)/\tau}{\sum_{k=1}^{V} \exp(l_{g,k} + n_k)/\tau},$$ (1)

where $\tau$ is a non-negative temperature, $n = -\log(-\log(u))$ and $u$ are uniform samples from $\mathcal{U}(0, 1)$. During the forward pass, codeword $i$ is chosen by $i = \text{argmax}_j p_{g,j}$ and in the backward pass, the true gradient of the Gumbel softmax outputs is used.

# 3 Training

To pre-train the model we mask a certain proportion of time steps in the latent feature encoder space (§ 3.1), similar to masked language modeling in BERT [9]. The training objective requires identifying the correct quantized latent audio representation in a set of distractors for each masked time step (§ 3.2) and the final model is fine-tuned on the labeled data (§ 3.3).

## 3.1 Masking

We mask a proportion of the feature encoder outputs, or time steps before feeding them to the context network and replace them with a trained feature vector shared between all masked time steps; we do not mask inputs to the quantization module. To mask the latent speech representations output by the encoder, we randomly sample without replacement a certain proportion $p$ of all time steps to be starting indices and then mask the subsequent $M$ consecutive time steps from every sampled index; spans may overlap.

## 3.2 Objective

During pre-training, we learn representations of speech audio by solving a contrastive task $\mathcal{L}_m$ which requires to identify the true quantized latent speech representation for a masked time step within a set of distractors. This is augmented by a codebook diversity loss $\mathcal{L}_d$ to encourage the model to use the codebook entries equally often.

$$\mathcal{L} = \mathcal{L}_m + \alpha \mathcal{L}_d$$ (2)

where $\alpha$ is a tuned hyperparameter.

**Contrastive Loss.** Given context network output $\mathbf{c}_t$ centered over masked time step $t$, the model needs to identify the true quantized latent speech representation $\mathbf{q}_t$ in a set of $K + 1$ quantized candidate representations $\tilde{\mathbf{q}} \in \mathbf{Q}_t$ which includes $\mathbf{q}_t$ and $K$ distractors [23, 54]. Distractors are uniformly sampled from other masked time steps of the same utterance. The loss is defined as

$$\mathcal{L}_m = -\log \frac{\exp(sim(\mathbf{c}_t, \mathbf{q}_t)/\kappa)}{\sum_{\tilde{\mathbf{q}} \sim \mathbf{Q}_t} \exp(sim(\mathbf{c}_t, \tilde{\mathbf{q}})/\kappa)}$$ (3)

where we compute the cosine similarity $sim(\mathbf{a}, \mathbf{b}) = \mathbf{a}^T \mathbf{b}/\|\mathbf{a}\|\|\mathbf{b}\|$ between context representations and quantized latent speech representations [19, 6].

**Diversity Loss.** The contrastive task depends on the codebook to represent both positive and negative examples and the diversity loss $\mathcal{L}_d$ is designed to increase the use of the quantized codebook representations [10]. We encourage the equal use of the $V$ entries in each of the $G$ codebooks by maximizing the entropy of the averaged softmax distribution $\mathbf{l}$ over the codebook entries for each

codebook $\bar{p}_g$ across a batch of utterances; the softmax disribution does not contain the gumbel noise nor a temperature:[2]

$$\mathcal{L}_d = \frac{1}{GV} \sum_{g=1}^{G} -H(\bar{p}_g) = \frac{1}{GV} \sum_{g=1}^{G} \sum_{v=1}^{V} \bar{p}_{g,v} \log \bar{p}_{g,v} \tag{4}$$

### 3.3 Fine-tuning

Pre-trained models are fine-tuned for speech recognition by adding a randomly initialized linear projection on top of the context network into $C$ classes representing the vocabulary of the task [4]. For Librispeech, we have 29 tokens for character targets plus a word boundary token. Models are optimized by minimizing a CTC loss [14] and we apply a modified version of SpecAugment [41] by masking to time-steps and channels during training which delays overfitting and significantly improves the final error rates, especially on the Libri-light subsets with few labeled examples.

## 4 Experimental Setup

### 4.1 Datasets

As unlabeled data we consider the Librispeech corpus [40] without transcriptions containing 960 hours of audio (LS-960) or the audio data from LibriVox (LV-60k). For the latter we follow the pre-processing of [27] resulting in 53.2k hours of audio. We fine-tune on five labeled data settings: 960 hours of transcribed Librispeech, the train-clean-100 subset comprising 100 hours (100 hours labeled), as well as the Libri-light limited resource training subsets originally extracted from Librispeech, these are train-10h (10 hours labeled), train-1h (1 hour labeled), train-10min (10 min labeled). We follow the evaluation protocol of Libri-light for these splits and evaluate on the standard Librispech dev-other/clean and test-clean/other sets.

We fine-tune the pre-trained models for phoneme recognition on the TIMIT dataset [13]. It contains five hours of audio recordings with detailed phoneme labels. We use the standard train, dev and test split and follow the standard protocol of collapsing phone labels to 39 classes.

### 4.2 Pre-training

Models are implemented in fairseq [39]. For masking, we sample $p = 0.065$ of all time-steps to be starting indices and mask the subsequent $M = 10$ time-steps. This results in approximately 49% of all time steps to be masked with a mean span length of 14.7, or 299ms (see Appendix A for more details on masking).

The feature encoder contains seven blocks and the temporal convolutions in each block have 512 channels with strides (5,2,2,2,2,2,2) and kernel widths (10,3,3,3,3,2,2). This results in an encoder output frequency of 49 hz with a stride of about 20ms between each sample, and a receptive field of 400 input samples or 25ms of audio. The convolutional layer modeling relative positional embeddings has kernel size 128 and 16 groups.

We experiment with two model configurations which use the same encoder architecture but differ in the Transformer setup: BASE contains 12 transformer blocks, model dimension 768, inner dimension (FFN) 3,072 and 8 attention heads. Batches are built by cropping 250k audio samples, or 15.6sec, from each example. Crops are batched together to not exceed 1.4m samples per GPU and we train on a total of 64 V100 GPUs for 1.6 days [38]; the total batch size is 1.6h.

The LARGE model contains 24 transformer blocks with model dimension 1,024, inner dimension 4,096 and 16 attention heads. We crop 320k audio samples, or 20sec, with a limit of 1.2m samples per GPU and train on 128 V100 GPUs over 2.3 days for Librispeech and 5.2 days for LibriVox; the total batch size is 2.7h. We use dropout 0.1 in the Transformer, at the output of the feature encoder and the input to the quantization module. Layers are dropped at a rate of 0.05 for BASE and 0.2 for LARGE [22, 12]; there is no layer drop for LV-60k.

We optimize with Adam [29], warming up the learning rate for the first 8% of updates to a peak of $5 \times 10^{-4}$ for BASE and $3 \times 10^{-4}$ for LARGE, and then linearly decay it. LARGE trains for 250k updates, BASE for 400k updates, and LARGE on LV-60k for 600k updates. We use weight $\alpha = 0.1$ for the diversity loss Equation 2. For the quantization module we use $G = 2$ and $V = 320$ for both models, resulting in a theoretical maximum of 102.4k codewords. Entries are of size $d/G = 128$ for BASE amd $d/G = 384$ for LARGE. The Gumbel softmax temperature $\tau$ is annealed from 2 to a minimum of 0.5 for BASE and 0.1 for LARGE by a factor of 0.999995 at every update. The temperature in the contrastive loss (Equation 3) is set to $\kappa = 0.1$. For the smaller Librispeech dataset, we regularize the model by applying an L2 penalty to the activations of the final layer of the feature encoder and scale down the gradients for the encoder by a factor of 10. We also use a slightly different encoder architecture where we do not use layer normalization, and instead of normalizing the raw waveform, the output of the first encoder layer is normalized. In the contrastive loss we use $K = 100$ distractors. We choose the training checkpoint with the lowest $\mathcal{L}_m$ on the validation set.

### 4.3 Fine-tuning

After pre-training we fine-tune the learned representations on labeled data and add a randomly initialized output layer on top of the Transformer to predict characters (Librispeech/Libri-light) or phonemes (TIMIT). For Libri-light, we train three seeds with two different learning rates (2e-5 and 3e-5) for all subsets and choose the configuration with lowest WER on dev-other subset decoded with the official 4-gram language model (LM) with beam 50 and fixed model weights (LM weight 2, word insertion penalty -1). For BASE on the labeled 960h subset we use a learning rate of 1e-4.

We optimize with Adam and a tri-state rate schedule where the learning rate is warmed up for the first 10% of updates, held constant for the next 40% and then linearly decayed for the remainder. BASE uses a batch size of 3.2m samples per GPU and we fine-tune on 8 GPUs, giving a total batch size of 1,600sec. LARGE batches 1.28m samples on each GPU and we fine-tune on 24 GPUs, resulting in an effective batch size of 1,920sec. For the first 10k updates only the output classifier is trained, after which the Transformer is also updated. The feature encoder is not trained during fine-tuning. We mask the feature encoder representations with a strategy similar to SpecAugment [41] detailed in Appendix B.

### 4.4 Language Models and Decoding

We consider two types of language models (LM): a 4-gram model and a Transformer [3] trained on the Librispeech LM corpus. The Transformer LM is identical to [51] and contains 20 blocks, model dimension 1,280, inner dimension 6,144 and 16 attention heads. We tune the weights of the language model (interval $[0, 5]$) and a word insertion penalty ($[-5, 5]$) via Bayesian optimization[3]: we run 128 trials with beam 500 for the 4-gram LM and beam 50 for the Transformer LM and choose the best set of weights according to performance on dev-other. Test performance is measured with beam 1,500 for the n-gram LM and beam 500 for the Transformer LM. We use the beam search decoder of [44].

## 5 Results

### 5.1 Low-Resource Labeled Data Evaluation

We first evaluate our pre-trained models in settings where the amount of labeled data is limited to get a sense of how the representations learned on unlabeled data can improve low resource settings. If a pre-trained model captures the structure of speech, then it should require few labeled examples to fine-tune it for speech recognition. The models are pre-trained on the audio data of either Librispeech (LS-960) or LibriVox (LV-60k) and most results are obtained by decoding with a Transformer language model (Transf.); Appendix C shows results with no language model at all as well as with an n-gram language model.

The LARGE model pre-trained on LV-60k and fine-tuned on only 10 minutes of labeled data achieves a word error rate of 5.2/8.6 on the Librispeech clean/other test sets. Ten minutes of labeled data corresponds to just 48 recordings with an average length of 12.5 seconds. This demonstrates that ultra-low resource speech recognition is possible with self-supervised learning on unlabeled data.

Table 1: WER on the Librispeech dev/test sets when training on the Libri-light low-resource labeled data setups of 10 min, 1 hour, 10 hours and the clean 100h subset of Librispeech. Models use either the audio of Librispeech (LS-960) or the larger LibriVox (LV-60k) as unlabeled data. We consider two model sizes: BASE (95m parameters) and LARGE (317m parameters). Prior work used 860 unlabeled hours (LS-860) but the total with labeled data is 960 hours and comparable to our setup.

| Model | Unlabeled data | LM | dev | | test | |
|---|---|---|---|---|---|---|
| | | | clean | other | clean | other |
| **10 min labeled** | | | | | | |
| Discrete BERT [4] | LS-960 | 4-gram | 15.7 | 24.1 | 16.3 | 25.2 |
| BASE | LS-960 | 4-gram | 8.9 | 15.7 | 9.1 | 15.6 |
| | | Transf. | 6.6 | 13.2 | 6.9 | 12.9 |
| LARGE | LS-960 | Transf. | 6.6 | 10.6 | 6.8 | 10.8 |
| | LV-60k | Transf. | 4.6 | 7.9 | 4.8 | 8.2 |
| **1h labeled** | | | | | | |
| Discrete BERT [4] | LS-960 | 4-gram | 8.5 | 16.4 | 9.0 | 17.6 |
| BASE | LS-960 | 4-gram | 5.0 | 10.8 | 5.5 | 11.3 |
| | | Transf. | 3.8 | 9.0 | 4.0 | 9.3 |
| LARGE | LS-960 | Transf. | 3.8 | 7.1 | 3.9 | 7.6 |
| | LV-60k | Transf. | 2.9 | 5.4 | 2.9 | 5.8 |
| **10h labeled** | | | | | | |
| Discrete BERT [4] | LS-960 | 4-gram | 5.3 | 13.2 | 5.9 | 14.1 |
| Iter. pseudo-labeling [58] | LS-960 | 4-gram+Transf. | 23.51 | 25.48 | 24.37 | 26.02 |
| | LV-60k | 4-gram+Transf. | 17.00 | 19.34 | 18.03 | 19.92 |
| BASE | LS-960 | 4-gram | 3.8 | 9.1 | 4.3 | 9.5 |
| | | Transf. | 2.9 | 7.4 | 3.2 | 7.8 |
| LARGE | LS-960 | Transf. | 2.9 | 5.7 | 3.2 | 6.1 |
| | LV-60k | Transf. | 2.4 | 4.8 | 2.6 | 4.9 |
| **100h labeled** | | | | | | |
| Hybrid DNN/HMM [34] | - | 4-gram | 5.0 | 19.5 | 5.8 | 18.6 |
| TTS data augm. [30] | - | LSTM | | | 4.3 | 13.5 |
| Discrete BERT [4] | LS-960 | 4-gram | 4.0 | 10.9 | 4.5 | 12.1 |
| Iter. pseudo-labeling [58] | LS-860 | 4-gram+Transf. | 4.98 | 7.97 | 5.59 | 8.95 |
| | LV-60k | 4-gram+Transf. | 3.19 | 6.14 | 3.72 | 7.11 |
| Noisy student [42] | LS-860 | LSTM | 3.9 | 8.8 | 4.2 | 8.6 |
| BASE | LS-960 | 4-gram | 2.7 | 7.9 | 3.4 | 8.0 |
| | | Transf. | 2.2 | 6.3 | 2.6 | 6.3 |
| LARGE | LS-960 | Transf. | 2.1 | 4.8 | 2.3 | 5.0 |
| | LV-60k | Transf. | 1.9 | 4.0 | 2.0 | 4.0 |

Our approach of jointly learning discrete units and contextualized representations clearly improves over previous work which learned quantized audio units in a separate step [4], reducing WER by a about a third.

A recent iterative self-training approach [42] represents the state of the art on the clean 100 hour subset of Librispeech but it requires multiple iterations of labeling, filtering, and re-training. Our approach is simpler: we pre-train on the unlabeled data and fine-tune on the labeled data. On the 100 hour subset of Librispeech, their method achieves WER 4.2/8.6 on test-clean/other which compares to WER 2.3/5.0 with the LARGE model in a like for like setup, a relative WER reduction of 45%/42%.

When the LARGE model uses an order of magnitude less labeled data (10h labeled), then it still achieves WER 3.2/6.1, an error reduction of 24%/29% relative to iterative self-training. Using only a single hour of labeled data, the same model achieves WER 3.9/7.6 which improves on both test-clean and test-other by 7%/12% - with two orders of magnitude less labeled data. We note that the Libri-

Table 2: WER on Librispeech when using all 960 hours of labeled data (cf. Table 1).

| Model | Unlabeled data | LM | dev | | test | |
|---|---|---|---|---|---|---|
| | | | clean | other | clean | other |
| **Supervised** | | | | | | |
| CTC Transf [51] | - | CLM+Transf. | 2.20 | 4.94 | 2.47 | 5.45 |
| S2S Transf. [51] | - | CLM+Transf. | 2.10 | 4.79 | 2.33 | 5.17 |
| Transf. Transducer [60] | - | Transf. | - | - | 2.0 | 4.6 |
| ContextNet [17] | - | LSTM | 1.9 | 3.9 | 1.9 | 4.1 |
| Conformer [15] | - | LSTM | 2.1 | 4.3 | 1.9 | 3.9 |
| **Semi-supervised** | | | | | | |
| CTC Transf. + PL [51] | LV-60k | CLM+Transf. | 2.10 | 4.79 | 2.33 | 4.54 |
| S2S Transf. + PL [51] | LV-60k | CLM+Transf. | 2.00 | 3.65 | 2.09 | 4.11 |
| Iter. pseudo-labeling [58] | LV-60k | 4-gram+Transf. | 1.85 | 3.26 | 2.10 | 4.01 |
| Noisy student [42] | LV-60k | LSTM | 1.6 | 3.4 | 1.7 | 3.4 |
| **This work** | | | | | | |
| LARGE - from scratch | - | Transf. | 1.7 | 4.3 | 2.1 | 4.6 |
| BASE | LS-960 | Transf. | 1.8 | 4.7 | 2.1 | 4.8 |
| LARGE | LS-960 | Transf. | 1.7 | 3.9 | 2.0 | 4.1 |
| | LV-60k | Transf. | 1.6 | 3.0 | 1.8 | 3.3 |

light data splits contain both clean and noisy data leading to better accuracy on test-other compared to test-clean. Increasing model size reduces WER on all setups with the largest improvements on test-other (BASE vs. LARGE both on LS-960) and increasing the amount of unlabeled training data also leads to large improvements (LARGE LS-960 vs. LV-60k).

## 5.2 High-Resource Labeled Data Evaluation on Librispeech

In this section we evaluate the performance when large quantities of labeled speech are available to assess the effectiveness of our approach in a high resource setup. Specifically, we fine-tune the same models as before on the full 960 hours of labeled Librispeech: BASE and LARGE pre-trained on LS-960 as well as LARGE pre-trained on LV-60k.

Table 2 shows that our approach achieves WER 1.8/3.3 on test-clean/other on the full Librispeech benchmark. This is despite a weaker baseline architecture: supervised training of our architecture achieves WER 2.1/4.6 (LARGE - from scratch) compared to WER 1.9/4.1 for ContextNet [17], the baseline architecture of the state of the art [42]. We use a simple Transformer with CTC which does not perform as well as seq2seq models [51].

Note that the vocabulary of our acoustic model (characters) does not match the vocabulary of the LM (words) which delays feedback from the LM and is likely to be detrimental. Most recent work [51, 58, 17, 42] uses the better performing word pieces [50] for both models. Moreover, our result is achieved without any data balancing such as [42]. Finally, self-training is likely complimentary to pre-training and their combination may yield even better results. Appendix E presents a detailed error analysis of our pre-trained models in various labeled data setups.

## 5.3 Phoneme Recognition on TIMIT

Next, we evaluate accuracy on TIMIT phoneme recognition by fine-tuning the pre-trained models on the labeled TIMIT training data. We fine-tune as for the 10 hour subset of Libri-light but do not use a language model. Table 3 shows that our approach can achieve a new state of the art on this dataset, reducing PER by a relative 23%/29% over the next best result on the dev/test sets. Appendix D shows an analysis of how the discrete latent speech representations related to phonemes. Other recent work on pre-training which evaluates on TIMIT includes [47] who solve multiple tasks to learn good representations of speech.

Table 3: TIMIT phoneme recognition accuracy in terms of phoneme error rate (PER).

|  | dev PER | test PER |
|---|---|---|
| CNN + TD-filterbanks [59] | 15.6 | 18.0 |
| PASE+ [47] | - | 17.2 |
| Li-GRU + fMLLR [46] | – | 14.9 |
| wav2vec [49] | 12.9 | 14.7 |
| vq-wav2vec [5] | 9.6 | 11.6 |
| **This work (no LM)** | | |
| LARGE (LS-960) | 7.4 | 8.3 |

Table 4: Average WER and standard deviation on combined dev-clean/other of Librispeech for three training seeds. We ablate quantizing the context network input and the targets in the contrastive loss.

|  | avg. WER | std. |
|---|---|---|
| Continuous inputs, quantized targets (Baseline) | 7.97 | 0.02 |
| Quantized inputs, quantized targets | 12.18 | 0.41 |
| Quantized inputs, continuous targets | 11.18 | 0.16 |
| Continuous inputs, continuous targets | 8.58 | 0.08 |

## 5.4 Ablations

A difference to previous work [5, 4] is that we quantize the latent audio representations only for the contrastive loss, i.e., when latents are used as *targets*, but not when the latents are *input* to the Transformer network. We motivate this choice by an ablating for which we adopt a reduced training setup to increase experimental turn around: we pre-train BASE on LS-960 for 250k updates with masking probability $p = 0.075$, fine-tune on train-10h for 60k updates on a single GPU with 640k samples per batch, or 40 sec of speech audio. We report the average WER and standard deviation on the concatenation of dev-clean and dev-other (dev PER) for three seeds of fine-tuning.

Table 4 shows that our strategy of continuous inputs with quantized targets (Baseline) performs best. Continuous latent speech representations retain more information to enable better context representations and quantizing the target representations leads to more robust training. Quantizing the latents both in the input and the targets performs least well, and explains the lower performance of prior work [5, 4]. Continuous targets reduce the effectiveness of self-supervised training since targets can capture detailed artifacts of the current sequence, e.g. speaker and background information, which make the task easier and prevent the model from learning general representations beneficial to speech recognition. The training accuracy of identifying the correct latent audio representation increases from 62% to 78.0% when switching from quantized to continuous targets. Continuous inputs and continuous targets perform second best but various attempts to improve it did not lead to better results (see Appendix F for this experiment and other ablations on various hyperparameters).

## 6   Conclusion

We presented wav2vec 2.0, a framework for self-supervised learning of speech representations which masks latent representations of the raw waveform and solves a contrastive task over quantized speech representations. Our experiments show the large potential of pre-training on unlabeled data for speech processing: when using only 10 minutes of labeled training data, or 48 recordings of 12.5 seconds on average, we achieve a WER of 4.8/8.2 on test-clean/other of Librispeech.

Our model achieves results which achieve a new state of the art on the full Librispeech benchmark for noisy speech. On the clean 100 hour Librispeech setup, wav2vec 2.0 outperforms the previous best result while using 100 times less labeled data. The approach is also effective when large amounts of labeled data are available. We expect performance gains by switching to a seq2seq architecture and a word piece vocabulary.

## Broader Impact

There are around 7,000 languages in the world and many more dialects. However, for most of them no speech recognition technology exists since current systems require hundreds or thousands of hours of labeled data which is hard to collect for most languages. We have shown that speech recognition models can be built with very small amounts of annotated data at very good accuracy. We hope our work will make speech recognition technology more broadly available to many more languages and dialects.

## Acknowledgments

We thank Tatiana Likhomanenko and Qiantong Xu for helpful discussion and their help with wav2letter integration.

## Footnotes

[1]Code and models are available at https://github.com/pytorch/fairseq

[2]Our implementation maximizes perplexity $\frac{GV - \sum_{g=1}^{G} \exp(-\sum_{v=1}^{V} p_{gv} \log p_{gv})}{GV}$ which is equivalent.

[3]https://github.com/facebook/Ax

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
