[Supplementary Material]

# Appendices

## A    Masking distribution

When choosing which time-steps to mask, each latent speech representation in an utterance is considered a candidate starting time-step with probability $p$ where $M$ is the length of each masked span starting from the respective time step; both are hyper-parameters. Sampled starting time steps are expanded to length $M$ and spans can overlap.

For a 15 sec long audio sample, the average mask length is 14.7 time-steps, corresponding to 299ms of audio, with a median of 10 time-steps, and a maximum of about 100 time steps; about 49% of all time-steps in the sample will be masked. A plot of the corresponding mask length distribution is shown in Figure 2 and an ablation of $M$ and $p$ as well as the effect of other masking strategies is shown in Table 5. Reducing $M$ results in increased prediction accuracy for the self-supervised but the task becomes trivial when spans with length one are masked, leading to poor performance on downstream speech recognition tasks. We also consider other masking strategies: w/o overlap uniform($a$,$b$) samples for each starting index a span length $M^s$ from interval $a$ to $b$ and masks the subsequent $M^s$ time-steps taking care not to overlap with existing spans; poisson($\lambda$) and normal($\mu$, $\sigma$) sample $M^s$ from Poisson and normal distributions.

Figure 2: Mask length distribution for a 15 second sample with $p = 0.065$ and $M = 10$.

Table 5: Ablations on settings for the masking strategy during pre-training. When masking without overlap, we choose starting time steps with $p = 0.037$ which results in the total number of masked tokens to match the baseline.

|  | avg WER | std |
|---|---|---|
| Baseline ($p = 0.075$) | 7.97 | 0.02 |
| Mask length $M = 8$ | 8.33 | 0.05 |
| Mask length $M = 12$ | 8.19 | 0.08 |
| Mask length $M = 15$ | 8.43 | 0.19 |
| Mask probability $p = 0.065$ | 7.95 | 0.08 |
| Mask probability $p = 0.06$ | 8.14 | 0.22 |
| Mask w/o overlap, uniform(1,31) | 8.39 | 0.02 |
| Mask w/o overlap, uniform(10,30) | 9.17 | 0.05 |
| Mask w/o overlap, poisson(15) | 8.13 | 0.04 |
| Mask w/o overlap, normal(15, 10) | 8.37 | 0.03 |
| Mask w/o overlap, length 10 | 9.15 | 0.02 |
| Mask w/o overlap, length 15 | 9.43 | 0.26 |

# B  Fine-tuning Setup

During fine-tuning we apply a masking strategy to the feature encoder outputs similar to SpecAugment [41]: we randomly choose a number of starting time steps for which a span of ten subsequent time-steps is replaced with a mask embedding; spans may overlap and we use the same masked time step embedding as during pre-training. We also mask channels by choosing a number of channels as starting indices and then expand each one to cover the subsequent 64 channels. Spans may overlap and the selected channel spans are set to zero value. We use LayerDrop [22, 12] at a rate of 0.05 for BASE and 0.1 for LARGE during fine-tuning.

Table 6 summarizes the fine-tuning hyper-parameter settings used for the different labeled data setup. Table 7 shows the decoding parameters used for final evaluations of the various labeled data setups for Librispeech pre-trained models and Table 8 shows decoding parameters for LibriVox.

Table 6: Fine-tuning hyperparameters

|  | timestep mask prob. | channel mask prob. | updates |
|---|---|---|---|
| 10 min | 0.075 | 0.008 | 12k |
| 1 hour | 0.075 | 0.004 | 13k |
| 10 hours | 0.065 | 0.004 | 20k |
| 100 hours | 0.05 | 0.008 | 50k |
| 960 hours | 0.05 | 0.0016 | 320k |
| TIMIT | 0.065 | 0.012 | 40k |

Table 7: Decoding parameters for Librispeech subsets for models pre-trained on Librispeech

|  | 4gram LM weight | 4gram word insert. | TransLM weight | TransLM word insert. |
|---|---|---|---|---|
| 10 min | 3.23 | -0.26 | 1.20 | -1.39 |
| 1 hour | 2.90 | -1.62 | 1.15 | -2.08 |
| 10 hours | 2.46 | -0.59 | 1.06 | -2.32 |
| 100 hours | 2.15 | -0.52 | 0.87 | -1.00 |
| 960 hours | 1.74 | 0.52 | 0.92 | -0.86 |

Table 8: Decoding parameters for Librispeech subsets for models pre-trained on Librivox.

|  | 4gram LM weight | 4gram word insert. | TransLM weight | TransLM word insert. |
|---|---|---|---|---|
| 10 min | 3.86 | -1.18 | 1.47 | -2.82 |
| 1 hour | 3.09 | -2.33 | 1.33 | -0.69 |
| 10 hours | 2.12 | -0.90 | 0.94 | -1.05 |
| 100 hours | 2.15 | -0.52 | 0.87 | -1.00 |
| 960 hours | 1.57 | -0.64 | 0.90 | -0.31 |

## C  Full results for Libri-light and Librispeech

Table 9: WER on the Librispeech dev/test sets when training on the Libri-light low-resource labeled data setups (cf. Table 1).

| Model | Unlabeled data | LM | dev | | test | |
|---|---|---|---|---|---|---|
| | | | clean | other | clean | other |
| **10 min labeled** | | | | | | |
| BASE | LS-960 | None | 46.1 | 51.5 | 46.9 | 50.9 |
| | | 4-gram | 8.9 | 15.7 | 9.1 | 15.6 |
| | | Transf. | 6.6 | 13.2 | 6.9 | 12.9 |
| LARGE | LS-960 | None | 43.0 | 46.3 | 43.5 | 45.3 |
| | | 4-gram | 8.6 | 12.9 | 8.9 | 13.1 |
| | | Transf. | 6.6 | 10.6 | 6.8 | 10.8 |
| LARGE | LV-60k | None | 38.3 | 41.0 | 40.2 | 38.7 |
| | | 4-gram | 6.3 | 9.8 | 6.6 | 10.3 |
| | | Transf. | 4.6 | 7.9 | 4.8 | 8.2 |
| **1h labeled** | | | | | | |
| BASE | LS-960 | None | 24.1 | 29.6 | 24.5 | 29.7 |
| | | 4-gram | 5.0 | 10.8 | 5.5 | 11.3 |
| | | Transf. | 3.8 | 9.0 | 4.0 | 9.3 |
| LARGE | LS-960 | None | 21.6 | 25.3 | 22.1 | 25.3 |
| | | 4-gram | 4.8 | 8.5 | 5.1 | 9.4 |
| | | Transf. | 3.8 | 7.1 | 3.9 | 7.6 |
| LARGE | LV-60k | None | 17.3 | 20.6 | 17.2 | 20.3 |
| | | 4-gram | 3.6 | 6.5 | 3.8 | 7.1 |
| | | Transf. | 2.9 | 5.4 | 2.9 | 5.8 |
| **10h labeled** | | | | | | |
| BASE | LS-960 | None | 10.9 | 17.4 | 11.1 | 17.6 |
| | | 4-gram | 3.8 | 9.1 | 4.3 | 9.5 |
| | | Transf. | 2.9 | 7.4 | 3.2 | 7.8 |
| LARGE | LS-960 | None | 8.1 | 12.0 | 8.0 | 12.1 |
| | | 4-gram | 3.4 | 6.9 | 3.8 | 7.3 |
| | | Transf. | 2.9 | 5.7 | 3.2 | 6.1 |
| LARGE | LV-60k | None | 6.3 | 9.8 | 6.3 | 10.0 |
| | | 4-gram | 2.6 | 5.5 | 3.0 | 5.8 |
| | | Transf. | 2.4 | 4.8 | 2.6 | 4.9 |
| **100h labeled** | | | | | | |
| BASE | LS-960 | None | 6.1 | 13.5 | 6.1 | 13.3 |
| | | 4-gram | 2.7 | 7.9 | 3.4 | 8.0 |
| | | Transf. | 2.2 | 6.3 | 2.6 | 6.3 |
| LARGE | LS-960 | None | 4.6 | 9.3 | 4.7 | 9.0 |
| | | 4-gram | 2.3 | 5.7 | 2.8 | 6.0 |
| | | Transf. | 2.1 | 4.8 | 2.3 | 5.0 |
| LARGE | LV-60k | None | 3.3 | 6.5 | 3.1 | 6.3 |
| | | 4-gram | 1.8 | 4.5 | 2.3 | 4.6 |
| | | Transf. | 1.9 | 4.0 | 2.0 | 4.0 |

Table 10: WER on Librispeech when using all 960 hours of Librispeech as labeled data (cf. Table 2).

| Model | Unlabeled data | LM | dev | | test | |
|---|---|---|---|---|---|---|
| | | | clean | other | clean | other |
| LARGE - from scratch | - | None | 2.8 | 7.6 | 3.0 | 7.7 |
| | - | 4-gram | 1.8 | 5.4 | 2.6 | 5.8 |
| | - | Transf. | 1.7 | 4.3 | 2.1 | 4.6 |
| BASE | LS-960 | None | 3.2 | 8.9 | 3.4 | 8.5 |
| | | 4-gram | 2.0 | 5.9 | 2.6 | 6.1 |
| | | Transf. | 1.8 | 4.7 | 2.1 | 4.8 |
| LARGE | LS-960 | None | 2.6 | 6.5 | 2.8 | 6.3 |
| | | 4-gram | 1.7 | 4.6 | 2.3 | 5.0 |
| | | Transf. | 1.7 | 3.9 | 2.0 | 4.1 |
| LARGE | LV-60k | None | 2.1 | 4.5 | 2.2 | 4.5 |
| | | 4-gram | 1.4 | 3.5 | 2.0 | 3.6 |
| | | Transf. | 1.6 | 3.0 | 1.8 | 3.3 |

## D  Analysis of Discrete Latent Speech Representations

Next, we investigate whether the discrete latent speech representations $\mathbf{q}_t$ learned by the quantizer relate to phonetic information: Using LARGE pre-trained on LV-60k and without any fine-tuning, we compute the discrete latents for the training data of TIMIT and compute the co-occurrence between human annotated phonemes and the latents. Ties are broken by choosing the phoneme which is most represented in the receptive field of $\mathbf{q}_t$. The training data contains 3696 utterances of average length 13.6 sec, or 563k discrete latents.

Figure 3 plots $P(phoneme|\mathbf{q}_t)$ and shows that many discrete latents appear to specialize in specific phonetic sounds. The silence phoneme (bcl) represents 22% of all human annotated speech data and is therefore also modeled by many different latents.

Figure 3: Visualization of the co-occurrence between discrete latent speech representations and phonemes. We plot the conditional probability $P(phoneme|\mathbf{q}_t)$ on TIMIT train data. The y-axis shows the collapsed 39 classes of phonemes and the x-axis is over the different discrete latents.

# E   Speech Recognition Error Analysis

In this section we study the most common errors our models make when fine-tuned on different amounts of labeled data (Table 11). We also show transcriptions of a few relatively challenging utterances from the dev-clean subset of Librispeech (Table 12).

We consider models with no lexicon or no language model decoding, marked None in Table 9: Larger capacity decreases error rates: LARGE on LS-960 improves the word error rate on dev-clean from 46.1 to 43 compared to BASE. Increasing the amount of unlabeled training data further decreases the error rate to 33.8 for LARGE on LS-960.

In the ten minute labeled data setup, the model is still able to recognize basic units of speech: Table 11 shows that most errors are around spelling of words, e.g., omitting silent characters such as *could → coud*, *know → now*, or ignoring repeated letters such as *still → stil*, *little → litle*. The LARGE LV-60k model achieves WER 38.3 on dev-clean and adding a Transformer language model enables to choose more likely pronunciations during the search and gives a large WER improvement to 5.0.

The ten minute models without lexicon and language model tend to spell words phonetically and omit repeated letters, e.g., *will → wil* (Table 11). Spelling errors decrease with more labeled data: with one hour of labeled data, slightly less common words move into the list of the most frequent errors, e.g., *heaven* and *food* are spelled phonetically. At ten hours, top errors include articles, e.g., *a*, *the* which are a common source of errors in speech recognition in general. There are also alternative spellings, *color* vs. *colour* as well as relatively rare words including person names, still spelled phonetically, e.g., *phoebe → feeby*.

At 100 hours, person names dominate the most frequent errors: *phoebe → phebe*, along with incorrect spacing *anyone → any one*, *awhile → a while*. Finally at 960 hours the word error rate falls to 2% and top errors are mostly articles, incorrect splits, and some very rare words or names such as *deucalion* or *gryce*.

The "from scratch" 960 hour model has a similar word error rate as the 100 hour pre-trained model and displays a similar pattern of errors.

The pre-trained speech representations can be easily adapted to recognize specific sounds while fine-tuning grounds these representations to the actual spelling.

Table 11: Top word errors for models trained on 10m, 1h and 10h, 100h, 960h of labeled data and decoded on the Librispeech dev-clean subset without a language model or lexicon (see Table 9 and Table 10 - None). In brackets is the total number of occurrences of each error.

| 10m LARGE LV-60k | 1h LARGE LV-60k | 10h LARGE LV-60k |
|---|---|---|
| all → al (181) | too → to (26) | in → and (15) |
| are → ar (115) | until → untill (24) | a → the (11) |
| will → wil (100) | new → knew (22) | o → oh (10) |
| you → yo (90) | door → dor (18) | and → in (9) |
| one → on (89) | says → sais (18) | mode → mod (9) |
| two → to (81) | soul → sol (17) | ursus → ersus (9) |
| well → wel (80) | bread → bred (16) | tom → tome (8) |
| been → ben (73) | poor → pore (16) | randal → randol (7) |
| upon → apon (73) | a → the (13) | the → a (7) |
| good → god (67) | either → ither (13) | color → colour (6) |
| see → se (66) | food → fud (13) | flour → flower (6) |
| we → whe (60) | doubt → dout (12) | phoebe → feeby (6) |
| little → litle (54) | earth → erth (12) | an → and (5) |
| great → grate (53) | led → lead (12) | cucumbers → cucombers (5) |
| your → yor (53) | sea → see (12) | egg → eg (5) |
| could → coud (51) | thee → the (12) | macklewain → macklewaine (5) |
| here → hear (51) | tom → tome (12) | magpie → magpi (5) |
| know → now (45) | add → ad (11) | milner → millner (5) |
| there → ther (45) | good → god (11) | stacy → staci (5) |
| three → thre (45) | heaven → heven (11) | trevelyan → trevellion (5) |
| still → stil (42) | mary → marry (11) | verloc → verlock (5) |
| off → of (40) | randal → randel (11) | ann → an (4) |
| don't → dont (37) | answered → ansered (10) | anyone → one (4) |
| shall → shal (36) | blood → blod (10) | apartment → appartment (4) |
| little → litl (35) | bozzle → bosel (10) | basin → bason (4) |

| 100h LARGE LV-60k | 960h LARGE LV-60k | 960h LARGE from scratch |
|---|---|---|
| a → the (13) | a → the (12) | and → in (20) |
| and → in (10) | and → in (9) | a → the (16) |
| in → and (10) | macklewain → mackelwaine (7) | in → and (13) |
| o → oh (8) | in → and (6) | the → a (10) |
| minnetaki → minnitaki (7) | o → oh (6) | in → an (8) |
| randal → randall (7) | bozzle → bosell (5) | and → an (5) |
| christie → cristy (6) | criss → chris (5) | clarke → clark (4) |
| macklewain → mackelwane (6) | bozzle → bosel (4) | grethel → gretel (4) |
| randal → randoll (6) | clarke → clark (4) | macklewain → mackelwaine (4) |
| bozzle → bosall (5) | colored → coloured (4) | this → the (4) |
| kaliko → calico (5) | grethel → gretel (4) | an → and (3) |
| trevelyan → trevelian (5) | lige → lyge (4) | anyone → one (3) |
| an → and (4) | the → a (4) | bozzle → basell (3) |
| and → an (4) | and → an (3) | buns → bunds (3) |
| anyone → one (4) | ann → marianne (3) | carrie → carry (3) |
| bozzle → bozall (4) | butte → bute (3) | criss → chris (3) |
| clarke → clark (4) | color → colour (3) | he's → is (3) |
| gryce → grice (4) | deucalion → ducalion (3) | his → is (3) |
| i'm → am (4) | forcemeat → meat (3) | honor → honour (3) |
| in → ind (4) | gryce → grice (3) | lattimer → latimer (3) |
| letty → lettie (4) | honor → honour (3) | millet → mellet (3) |
| phoebe → phebe (4) | kearny → kirney (3) | pyncheon → pension (3) |
| the → a (4) | nuova → noiva (3) | tad → ted (3) |
| ann → anne (3) | thing → anything (3) | thing → anything (3) |
| awhile → while (3) | this → the (3) | trevelyan → trevelian (3) |

Table 12: Examples of transcription of selected utterances from the dev-clean subset by various models without a language model or lexicon. Capitalized words indicate errors.

| Model | Transcription |
|---|---|
| Reference | i'm mister christopher from london |
| 10m LV-60k | IM mister CRESTIFER FROME LUNDEN |
| 1h LV-60k | IM mister CRISTIFFHER from LOUNDEN |
| 10h LV-60k | i'm mister CHRYSTEPHER from london |
| 100h LV-60k | i'm mister christopher from london |
| 960h LV-60k | i'm mister christopher from london |
| 960h scratch | I MISSTER christopher from london |
| Reference | il popolo e una bestia |
| 10m LV-60k | ILPOPULAR ONABESTIA |
| 1h LV-60k | O POPOLAONABASTIA |
| 10h LV-60k | U POPULAONABASTIAR |
| 100h LV-60k | O POPALOON A BASTYA |
| 960h LV-60k | YOU'LL POP A LAWYE ON A BAISTYE |
| 960h scratch | OL POPALOY ON ABESTIA |
| Reference | he smelt the nutty aroma of the spirit |
| 10m LV-60k | he SMELTD the NUDY aroma of the spirit |
| 1h LV-60k | he SMELTD the NUDDY ARROMA of the spirit |
| 10h LV-60k | he smelt the NUDDY ERROMA of the spirit |
| 100h LV-60k | he smelt the NUDDY aroma of the spirit |
| 960h LV-60k | he smelt the NUTTIE aroma of the spirit |
| 960h scratch | he smelt the nutty EROMA of the spirit |
| Reference | phoebe merely glanced at it and gave it back |
| 10m LV-60k | FEABY MEARLY glanced at it and gave it BAK |
| 1h LV-60k | FIEABY merely glanced at it and gave it back |
| 10h LV-60k | FEEBY merely glanced at it and gave it back |
| 100h LV-60k | BEBE merely glanced at it and gave it back |
| 960h LV-60k | phoebe merely glanced at it and gave it back |
| 960h scratch | phoebe merely glanced at it and gave it back |
| Reference | sauterne is a white bordeaux a strong luscious wine the best known varieties being |
| 10m LV-60k | SULTERIN is a white BORDOE a strong LUCHOUS WIN the best NOWN VERIATYS being |
| 1h LV-60k | CLTEREN is a white BORDO a strong LUCHIOUS wine the best known VERIETIES being |
| 10h LV-60k | SOTERN is a white BOURDO a strong LUCIOUS wine the best known VORIETIES being |
| 100h LV-60k | SOTERN is a white BORDAUX a strong LUCIOUS wine the best known varieties being |
| 960h LV-60k | SOTERN is a white bordeaux a strong luscious wine the best known varieties being |
| 960h scratch | SOTERAN is a white bordeaux a strong luscious wine the best known varieties being |
| Reference | i happen to have mac connell's box for tonight or there'd be no chance of our getting places |
| 10m LV-60k | i HAPEND to have MECONALES BOXS for TONIT ORE THIRLD be no chance of OR GETING places |
| 1h LV-60k | i happen to have MACCONNEL'S BOCXS for tonight or TE'ELD be no chance of our getting places |
| 10h LV-60k | i HAPPENED to have MUKONNEL'S box for tonight or THERED be no chance of our getting places |
| 100h LV-60k | i HAPPENED to have MC CONNEL'S box for TO NIGHT or there'd be no chance of our getting places |
| 960h LV-60k | i happen to have MC CONALL'S box for TO NIGHT or there'd be no chance of our getting places |
| 960h scratch | i HAPPENE to have MACONEL'S box for TO NIGHT or there'd be no chance of our getting places |

# F   Ablations

Table 13 ablates various hyperparameter choices of our architecture. The setup for the baseline model is described in § 5.4. First, we tried to improve the continuous input and continuous target model (§ 5.4) by adding an MLP on top of the continuous target representation and we also tried to use a separate set of encoder parameters for the representations used as input and targets (Separate encoders). Both did not lead to meaningful improvements.

Increasing the receptive field size from 25ms to 30ms had little effect. Setting the diversity penalty weight ($\alpha$) too low results in lower codebook usage and lower performance. Setting it too high leads to slight instability. Doubling the number of relative positional embeddings to 256 also did not help. Stopping gradients from the quantizer to the encoder shows that the encoder requires training signal from the quantizer as well.

Next, increasing the number of negatives did not result in better performance ($K = 200$) and sampling negatives from the entire batch of utterances hurt performance, likely because candidates from other utterances are easy to distinguish. Sampling negatives from any time step in the utterance, masked or unmasked, does not help and is more computationally expensive. Gumbel noise is important and increasing the number of codebooks did not result in better performance.

Table 13: Ablation of various hyper-parmeter choices. We report average WER and standard deviation on combined dev-clean/other of Librispeech for three seeds of training.

| | avg. WER | std. |
|---|---|---|
| Baseline ($p = 0.075$, $\alpha = 0.1$) | 7.97 | 0.02 |
| Continuous inputs, continuous targets | 8.58 | 0.08 |
| + MLP on targets | 8.51 | 0.05 |
| + Separate encoders | 8.90 | 0.01 |
| receptive field 30ms | 7.99 | 0.06 |
| diversity penalty | | |
| $\alpha = 0$ | 8.48 | 0.08 |
| $\alpha = 0.05$ | 8.34 | 0.08 |
| $\alpha = 0.2$ | 8.58 | 0.45 |
| Conv pos emb, kernel 256 | 8.14 | 0.05 |
| No gradient to encoder from quantizer | 8.41 | 0.08 |
| Negatives | | |
| $K = 200$ same utterance | 8.12 | 0.05 |
| $K = 50$ same utterance + $K = 50$ from batch | 8.79 | 0.06 |
| Sample negatives from any time step | 8.07 | 0.02 |
| No Gumbel noise | 8.73 | 0.42 |
| Codebook | | |
| G=4, V=18 | 9.02 | 0.38 |
| G=8, V=8 | 8.13 | 0.07 |
| Predict exactly $U$ time steps from edges | | |
| $U = 1$ | 9.53 | 0.91 |
| $U = 5$ | 8.19 | 0.07 |
| $U = 10$ | 8.07 | 0.07 |
| $U = 15$ | 7.89 | 0.10 |
| $U = 20$ | 7.90 | 0.01 |

We also investigated predicting only time steps immediately next to the last unmasked time step for each span. This enables to better control the difficulty of the pre-training task. Given the leftmost or rightmost unmasked time step next to a masked span, we compute the contrastive loss only for the first $U$ masked time steps next to these unmasked spans. Predicting only up to one time step performs poorly because there is little training signal from each utterance and predicting more time steps performs better but does not significantly outperform predicting all masked time steps. Increasing the number of training updates helps but this increases training time.