[Reviews · NeurIPS 2020]

Review 1

Summary and Contributions: This paper studies self-supervised masked prediction as a pre-training task for speech recognition under resource constrained scenarios. It closely follows prior work [4, 5] in its approach. [4, 5] utilized a two-step training procedure in which neural network quantizers first produced a discrete representation of the speech, and in the second stop these discrete embeddings were used to learn contextualized representations with a masked prediction objective. This paper differentiates itself from [4, 5] by training the whole model end-to-end. After pre-training, the model is fine-tuned for ASR (word recognition on Librispeech/Libri-light, phone recognition on TIMIT) by appending a lightweight output layer to predict phones/characters which is then trained with a CTC objective. SoTA results for phone recognition on TIMIT are reported. For the Librispeech/Libri-light experiments, results are reported for a spectrum of scenarios representing 10 minutes of transcribed speech available for fine-tuning all the way up through the full 960 hour Librispeech training set. Under the very low-resource scenarios, the paper reports SoTA WERs by a large margin. When using the full 960-hour training set, the proposed method achieves performance comparable to the current SoTA.

Strengths: The paper demonstrates very strong performance on several widely-used speech recognition benchmark datasets (Librispeech, TIMIT) under extremely resource-constrained settings. For example, on Librispeech, the proposed method achieves WERs with only 1 hour of labelled speech that are comparable to contemporary state of the art approaches that use 100 hours of labelled speech. In the case of TIMIT, the paper establishes a new SoTA for phone recognition.

Weaknesses: The main weakness I see in this paper is that while the experiments focus on a low-resource condition for acoustic model training, the full LibriSpeech language model training dataset appears to have been used. The WERs achieved are impressive when very limited amounts of transcribed speech are used (10 min, 1 hour, 10 hours), but when the official LibriSpeech LM (trained on the text from ~14,500 audiobooks) is incorporated into decoding, it is not clear whether the experiments still represent a realistic low-resource scenario. It would have been very interesting to see how the proposed method performs when the language model is similarly faced with a limited amount of training data. In line 164 there is a detail missing regarding what units are predicted by the output layer for Librispeech/Libri-light (I assume it is characters)

Correctness: Aside from the methodological criticism I raised in the "weaknesses" section, the claims and methods of the paper are solid.

Clarity: The paper is well written.

Relation to Prior Work: The paper's approach is very similar to that of [4, 5] but with some important differences that are adequately highlighted by the authors.

Reproducibility: Yes

Additional Feedback: Post-rebuttal feedback: Thank you to the authors for submitting a thoughtful rebuttal. My view of this paper is still positive and I will leave my score unchanged.


Review 2

Summary and Contributions: This paper realizes self-supervised training of speech signals and applied the pre-trained network to ASR tasks by fine-tuning, which is quite similar to the methodology developed in NLP. The technical novelty of this paper is that performing two-stage optimization of the discretization of audio signals and contextual modeling based on masking is now optimized in an end-to-end manner. With several elaborations. the method successfully trains pre-trained networks. The most notable contribution of this paper is that it shows the significant effect of this methodology in the semi-supervised ASR setup. The method archives reasonable ASR performance only with 10 min. of the supervised data. The method also archives the state-of-the-art performance in both Librispeech and TIMIT tasks. This method would be a breakthrough contribution to speech processing like pre-training based methods show the breakthrough in NLP.

Strengths: - novel end-to-end self-supervised training of speech signals. The two stage optimization already exists but the paper realizes an end-to-end optimization of these two stage methods. - shows the state-of-the-art peformance. - it could be reproducible since the paper carefully describes hyper-parameters, will release the open source version of the proposed method, and all experiments are based on the public database.

Weaknesses: - similar to the other pre-training method like BERT, this requires massive computational resources. It is not easy to reproduce the result in this computational perspectives. - survey of the paper is not enough (or at least less structured). - frankly, the paper is difficult to read. We need to check the references to understand detailed descriptions, and it is not very self-sufficient.

Correctness: - the claims of this method look correct overall. - one of my concerns is that the contrastive loss is based on the internal representation of the first discretization stage, which is also learned, i.e., all values appeared in the contrastive loss functions are variables. I feel that this makes training unstable. Probably, the other loss functions and careful optimization solves such an unstable learning process, but I'm not very convinced that this training works well conceptually.

Clarity: - sometimes, the detailed explanation of the variables in the math notation is missing, e.g., what is $T$? - frankly, I have difficulties reading this paper because the paper uses citations instead of explaining the details. So, the paper lacks self-consistency. I think it comes from a page size limitation and instead, the paper tries to describe a lot of results and some tweaks for the implementation. So, I think it is acceptable.

Relation to Prior Work: - Recent cycle-consistency and speech chain based semi-supervised models are also related. They use ASR to convert audio to discrete representation and TTS to get the reconstruction loss. The discretization concept is the same. Especially cycle-consistency training below tries to train ASR/TTS in an end-to-end manner, and this direction should be discussed as a related study. - Hori, Takaaki, et al. "Cycle-consistency training for end-to-end speech recognition." ICASSP 2019-2019 IEEE International Conference on Acoustics, Speech and Signal Processing (ICASSP). IEEE, 2019. - Tjandra, Andros, Sakriani Sakti, and Satoshi Nakamura. "Listening while speaking: Speech chain by deep learning." 2017 IEEE Automatic Speech Recognition and Understanding Workshop (ASRU). IEEE, 2017. - As I mention below, ASR actually has a lot of pair of data. So, the paper should also discuss supervised pre-training methods in the ASR application context. (Also, image processing has similar trends. Everyone uses supervised pre-trained models) - There are a lot of pre-training based classical speech recognition techniques (transfer learning, teacher-student learning), which should be discussed in the paper. - The other well-known method is a bottleneck feature. - Grézl, Frantisek, et al. "Probabilistic and bottle-neck features for LVCSR of meetings." 2007 IEEE International Conference on Acoustics, Speech and Signal Processing-ICASSP'07. Vol. 4. IEEE, 2007.

Reproducibility: Yes

Additional Feedback: - for ASR, we actually have a lot of transcribed data, and we can make first make a very strong ASR model and perform transfer learning. Therefore, the result of this paper is impressive, but it cannot be an essential technique compared with NLP pre-trained models. I suggest the authors apply this method to low-resource ASR tasks. - I'd like to have some clarifications on how to extract $K$ detractors. Is it random or try to exclude the codebook corresponding to q_t?


Review 3

Summary and Contributions: This paper proposes an unsupervised pre-training approach for speech recognition. The paper combines recent ideas in unsupervised pre-training and discrete representation learning into an end-to-end approach. The approach is validated on Librispeech and TIMIT tasks.

Strengths: Soundness of the claims --------------------------------- This paper provides a very good argument why optimal approaches to unsupervised pre-training is an important research area. The proposed approach is technically sound and has been evaluated on two tasks. Significance ---------------- This paper will be of interest to a large community (speech recognition and other fields). Novelty ---------- The proposed solution combines well known techniques in a novel way.

Weaknesses: Soundness of the claims --------------------------------- The paper reads as a lab report that provides limited argumentation for why many of the design decision have been made. Significance ---------------- The significance of this paper is limited by the choice of the tasks. The paper would have been significantly different in terms of quality had you applied you approach to some standard semi-supervised learning tasks in ASR. Such tasks have a number of very strong baselines and are very familiar to most of speech recognition community. Novelty ---------- You have stated main weakness of the novelty of your paper in the paper.

Correctness: I believe the claims made in this paper and the empirical methodology to be correct.

Clarity: This paper is written as a lab report which offers a very limited analysis and discussion about options available and decisions made.

Relation to Prior Work: I believe the prior work that enabled you to build the proposed solution have been adequately referenced. Other aspects, such as options available for discretising continuous representations, I believe have not been adequately referenced.

Reproducibility: Yes

Additional Feedback: I have read through other reviews and authors' rebuttal. I respectfully disagree with authors rebuttal. In particular, (a) This proposal does read like a lab report with limited amount of justification provided. Please move "we use a convolutional layer with kernel size 128 and 16 groups similar to", "we randomly sample without replacement p = 0.065 of all time steps to be starting indices and then mask the subsequent M = 10 consecutive time steps from every sampled index"-like sentences as much as possible to section 4 where they belong. Otherwise how 4 is different from 2 and 3. (b) This proposal does not use either the largest or most representative for your task data sets in speech recognition community. (Please search for semi-supervised training outside of NeurIPS/ICLR/ICML)


Review 4

Summary and Contributions: This study looks at the problem of learning representations from audio, when under labeling limitations. The proposed approach features two changes from traditional approaches: using continuous latent speech representations (quantizing only for training), and a simple two-step training regime (train on unlabelled data, fine-tune on labeled data). The method is evaluated on the Librispeech and TIMIT datasets. When looking at performance as the about of labeled data is reduced, the proposed method performs much better than pre-existing work. The method is also demonstrated to perform comparably to pre-existing work on all labeled data. Finally, the authors provide an ablation experiment demonstrating the benefit of keeping latent speech representations continuous.

Strengths: The major strengths of this work are straight-forward and reasonable methodological changes that this paper implements, alongside experiments that directly measure their benefit. The end result is a shift towards neural network architectures that are less complicated, and easier to train -- while performing better. Furthermore, the authors provide a solid ablation study and comparison with competing methods. The proposed method is evaluated with different language models and different neural network capacities. These ablations allow us to help factor out confounding reasons for increases in performance. Furthermore, when looking at restricting labeled data, there is a trend of the proposed method continuing to perform at a higher margin better than the baseline as labels are reduced.

Weaknesses: A weakness of this work is that from this work alone it’s not clear why the proposed changes should work well for the problem domain. Moreover, why the interaction of the two proposed changes is so beneficial. While this is a problem in the body of work that effectively searches through the neural-network-architecture space, it would be very beneficial to try and focus on justifying more rigorously the design choices made. An example of how this could be done is designing a toy problem that exemplifies that pre-existing work cannot handle this case, and such, the proposed changes should be accepted. As a result of this, it’s not clear how significant this work is/will be. Minor comments: It would be useful to explore/discuss why the choice not to quantize is beneficial to the limited labeled data setting. It’s not necessarily clear if this method is successful for its two-phase training regime. This method could trivially be extended so that it could iteratively apply it’s two stages. I would be curious if this further improved performance. Experiment 1: Why is “Discrete BERT” the only baseline that is evaluated in the limited regimes? Please justify this decision and/or include the performance of the other methods (possibly in the appendix). Experiment 2: Why are the methods featured in “Experiment 1” not also all included in this experiment? The “Broader Impact” discussion could be improved instead of discussing speech recognition at a high level, focusing instead on how the choices introduced here may also benefit/harm the broader community. For example, a pre-training fine-tuning regime may strongly bias models towards the data used for fine-tuning, which could introduce errors/issues.

Correctness: The claims and methods appear correct.

Clarity: The paper is not clearly written. Here are some actionable items that could greatly improve the clarity of this work: - Bring the justification of quantization from section 5.3 into the introduction (to not constrain information flow). It's not clear to the reader for most of the paper what the proposed changes are, and why they are reasonable choices. - Move the discussion of how the model compares to pre-existing works into a single section. This allows the discussion about related work that is dispersed throughout the paper to be condensed into a single more focused discussion. It also has the added benefit of not potentially distracting the reader. - Run a spell-checker on the text (e.g., Line 210, Line 224).

Relation to Prior Work: There exists a discussion about the differences between this work and previous works; however, they’re intermixed throughout the paper.

Reproducibility: Yes

Additional Feedback: The appendix contains lots of great experiments and analysis. I would strongly encourage referring more to these results within the paper.

[Author Response · NeurIPS 2020]

We thank the reviewers for their fruitful comments!

**Response to Reviewer 2:**   We predict characters for Librispeech/Libri-light. Thank you for the pointer!

*"when the official LibriSpeech LM ... is incorporated into decoding, it is not clear whether the experiments still represent*
*a realistic low-resource scenario."* - Please see Appendix C for results without any language model. For the main paper,
we show results with the entire LM data similar to other recent work on low resource ASR (e.g., Park et al., 2020).

**Response to Reviewer 3:**   Thank you for the pointers to related work, we will consider discussing them in the next
version of the paper. We will also try to make it more self-contained given the space restrictions.

*"I'm not convinced that this training works well conceptually."* - The joint training poses challenges which we successfully
tackled. It enables the model to learn units that are useful to solve the contrastive task instead of having to work with
fixed units that may not be optimal. Moreover, in computer vision, similar approaches are achieving the best results for
pre-training ("Unsupervised Learning of Visual Features by Contrasting Cluster Assignments. Caron et al., 2020).

*"... for ASR, we have a lot of transcribed data, and we can make a strong ASR model and perform transfer learning."* -
We respectfully disagree: while this statement is true for English and possibly a few other languages, the vast majority
of languages have very little if any transcribed data. Moreover, our approach outperforms the best supervised models,
even when training on all available transcribed data of Librispeech.

*"... how to extract $K$ detractors."* - The distractors are quantized latent speech representations sampled from masked
time-steps. If another masked time-step uses the same quantized latent, then it won't be sampled.

**Response to Reviewer 4:**   We respectfully disagree that many design choices have not been justified: we provide a
through evaluation of why quantizing targets is a good choice (see Sec 5.4). We compare our joint approach (quantization
and context representation learning) to a pipelined approach (Discrete BERT). We motivate the diversity loss and why
the encoder needs to be stabilized. Many other design choices such as gumbel softmax for quantization and the encoder
network architecture have been evaluated in previous work.

*"The paper would have been significantly different in terms of quality had you applied you approach to some standard*
*semi-supervised learning tasks."* - We focused on one of the largest datasets in the speech recognition community to
demonstrate the effectiveness of the approach when large amounts of labeled as well as unlabeled data is available.
This follows other recent work on semi-supervised methods for speech such as "Improved Noisy Student Training
for Automatic Speech Recognition. Park et al., 2020" and "End-to-end ASR: from Supervised to Semi-Supervised
Learning with Modern Architectures. Synnaeve et al., 2020" which achieve some of the strongest results.

**Response to Reviewer 5:**   *"... from this work alone it's not clear why the proposed changes should work well for the*
*problem domain. Moreover, why the interaction of the two proposed changes is so beneficial."* - The major changes to
prior work are: (1) quantize only targets, (2) jointly learn the quantization and the contextualized representations as
well scaling the model and data. (1) is thoroughly ablated in Sec 5.3. Our intuition is that discretized representations
are more robust to artifacts in the data that generalize less well. This is useful for the learning task in the loss function
(targets) but not when we want to build rich context representations (inputs). (2) is the major change to [1] "Discrete
BERT" which is outperformed by our joint approach (see Table 1). This shows that joint training works better than
pipelining discretization and context representation learning. The former enables adjusting the discretization when
needed while the latter has to work with a fixed discretization which is less flexible.

Overall, our design choices are highly effective: wav2vec 2.0 outperforms the best other semi-supervised methods by a
large margin on 100h labeled data and shows comparable results to the state of the art on 960h labeled data.

*"Experiment 1: Why is Discrete BERT the only baseline that is evaluated in the limited regimes?"* For the very low
resource setups (10min, 1h, 10h), this is the only competitive baseline, the only other model is reported in the original
Libri-light paper with far higher WER than Discrete BERT or our model (WER 92% (10min), 64% (1h), 44% (10h)).

*"Experiment 2: Why are the methods featured in "Experiment 1" not also all included in this experiment?"* - I believe
you are referring to the 960h labeled data setup. Previous work simply did not report results for this high resource setup.

*"It's not necessarily clear if this method is successful for its two-phase training regime. This method could trivially be*
*extended so that it could iteratively apply it's two stages. I would be curious if this further improved performance."*

Pre-training followed by fine-tuning is not an innovation of this paper. It has been previously applied to ASR in
"Effectiveness of self-supervised pre-training for speech recognition". Baevski et al., 2020. Once the model is fine-tuned
(second stage), pre-training on unlabeled data again is unlikely to benefit the model. Note that the first stage does not
use any labeled data.

[Meta-Review · NeurIPS 2020]

This paper proposes an end-to-end self-supervised learning approach for speech representations. It can serve as the unsupervised pre-training for fast and robust deployment of automatic speech recognition systems, especially for those with low resource or limited amounts of labeled data. The authors reported compelling performance of the proposed technique on Librispeech and TIMIT. This is a strong paper and all reviewers are supportive for acceptance. Large-scale unsupervised pre-training has made great impacts in vision and NLP, the work reported here is analogous in the speech community in that effort. That being said, there are still minor concerns raised by reviewers in the review and discussion. For instance, the exposition can be further polished in the final version to make it more accessible to the readers.